# Effect of Temperature on Passive Film Characteristics of LPBF (Laser Powder-Bed Fusion) Processing on UNS-S31603

**DOI:** 10.3390/ma17143420

**Published:** 2024-07-11

**Authors:** Reece Goldsberry, Deeparekha Narayanan, Raymundo Case, Bilal Mansoor, Homero Castaneda

**Affiliations:** National Corrosion and Materials Reliability Laboratory, Department of Materials Science and Engineering, Texas A&M University, 1041 Rellis Pkwy, Bryan, TX 77807, USA; reece1579@tamu.edu (R.G.); ndeeparekha@tamu.edu (D.N.); raymundo.case@tamu.edu (R.C.); bilal.mansoor@tamu.edu (B.M.)

**Keywords:** LPBF 316L, temperature study, pitting corrosion, passive films

## Abstract

The effect of temperature on the localized corrosion resistance and passive film characteristics of laser powder-bed fusion (LPBF) 316L (UNS S31603) was studied in a buffered 3.5 wt% NaCl solution at 25, 50, and 75 °C. DC techniques such as cyclic potentiodynamic polarization showed lower passive current densities, high breakdown potentials, and a higher resistance to initial breakdown compared with wrought 316L samples at all temperatures. However, LPBF 316L was more susceptible to metastable pitting at potentials before film breakdown and higher damage accumulation post film breakdown. AC techniques, such as Mott–Schottky analysis and electrochemical impedance spectroscopy, showed that the formed passive film was more robust on the LPBF 316L samples at all temperatures, accounting for the higher initial resistance to pitting. However, with increasing temperatures, the film formed had an increasing concentration of defect density. Passive compositions at the various test temperatures studied using X-ray photoelectron spectroscopy (XPS) showed that the LPBF samples showed higher amounts of Cr and Fe oxides and hydroxides compared with the wrought samples, which made the passive films on the LPBF samples more compact and protective. Investigation of the pits formed on the LPBF showed the preferential regions of attack were the melt-pool boundaries and cell interiors due to their being depleted of Cr and Mo when compared with the boundaries and matrix.

## 1. Introduction

Additive manufacturing (AM) is a near-net-shape manufacturing process where components are built with a layer-by-layer additive process. AM processes can be used to manufacture components from a variety of metals, mixed metal composites, and non-metallics [1,2,3]. Two main advantages of AM are the production of prototypes/end-use products with shorter lead times and the ability to produce geometrically complex components in a single step without secondary processing. Implementing weight reduction through AM can significantly impact the fuel consumption of the automotive, aerospace, and railway industries [4]. Some of the most common AM methods used for producing metallic components are based on laser, binder jetting, arc welding, and electron beam technologies [5].

Type 316L stainless steel is heavily used in various industries because it can provide decent mechanical properties, good corrosion resistance, and biocompatibility at a relatively low price point [6,7,8]. Type 316L owes its corrosion resistance to forming a compact chromium oxide layer that protects the base material from the surrounding environment [7,8,9]. These characteristics of 316L stainless steel make it one of the most studied and used steel alloy systems in the additive manufacturing of metallic components [1,10,11,12]. Laser powder-bed fusion (LPBF) is a unique AM method that subjects the material to high localized melting and solidification rates, resulting in a complex and anisotropic microstructure in the as-printed condition that differs from conventionally produced 316L components. The microstructure of LPBF components is characterized by a heterogenous microstructure consisting of various cell morphologies depending on the temperature gradients and heat transfer directions at the various areas of the powder bed [13,14,15]. Care must be taken when selecting processing parameters to ensure the components can be produced without defects. The most common defects in LPBF components are the presence of porosity/voids, unmelted powder, lack of fusion, and detrimental microstructural features. Optimization of specific parameters such as laser power, scan speed, hatch distance, layer height, and layer orientation allows components to be produced with little to no defects.

AM 316L has been shown to have increased corrosion resistance compared with its wrought counterpart in a range of environments [12,16,17]. This increase in the corrosion resistance of 316L produced through LPBF methods is attributed to two predominant theories. Chao et al. showed that the rapid melting and solidification of the 316L powder did not allow the formation of the MnS particles, showing a marked increase in the pitting potential measured through cyclic potentiodynamic polarization compared with wrought 316L [17]. MnS inclusions in traditionally-produced 316L microstructures have been shown to have a negative impact on the overall corrosion resistance due to its anodic nature in comparison with the matrix [18]. Second, the high density of cell boundaries due to the sub-grain cellular type structure allows the LPBF 316L to produce a thicker and more compact passive layer that provides more protection than the passive layer formed on wrought 316L [12,16].

In this work, the effect of temperature on the pitting corrosion resistance of LPBF 316L stainless steel was studied and compared with its wrought counterpart. Scanning electron microscopy (SEM) was used to characterize the microstructures developed, and a combination of DC electrochemical techniques such as cyclic polarization, to study the localized corrosion response, as well as AC techniques such as electrochemical impedance spectroscopy (EIS) and Mott–Schottky, to study characteristics of the passive film formed, were used.

Electrochemical testing was performed at 25, 50, and 75 °C to study the differences in electrochemical behavior at ambient as well as elevated temperatures. Immersion tests were performed at the studied temperatures followed by high-resolution surface characterization to provide insights into the chemical composition of the passive film formed. SEM imaging was used to capture the differences in pit morphologies in the wrought and LPBF samples after CPP tests at the three test temperatures. Through these tests, the aim was to deepen the understanding of the influence of temperature on the passivation characteristics of LPBF 316L to understand if the well-documented superior corrosion resistance under ambient conditions was observed at elevated temperatures as well.

## 2. Materials and Methods

The 316L LPBF samples were produced with a UNS31603 powder (Powder Alloy Corporation, Loveland, OH, USA) with a powder diameter of 45 μm ± 15 μm. The LPBF samples were printed with an EOS M 100 DMLS printer (EOS, Munich, Germany). Applied energy density was 88.5 J/mm^3^ with an applied laser power of 170 W, scan speed 800 mm/s, hatch spacing 120 µm, and a layer height of 20 µm. The nominal composition of the 316L stainless steel powder and wrought material were both in accordance with ASTM A666 [19] The average relative composition of the LPBF and wrought 316L materials measured through energy dispersive X-ray spectroscopy (EDS) is shown in Table 1. EDS was performed using the Oxford instruments detector equipped in a TESCAN FERA3 scanning electron microscope (SEM) (TESCAN, Brno, Czech Republic). The parameters used to capture images and perform EDS scans were an accelerating voltage of 20 kV and a working distance of ~9 mm. The relative EDS composition was obtained by acquiring point scans at multiple points over a view field of about 180 μm to acquire a good approximation of relative composition of heavy elements in samples. Lighter elements such as C, O, and Si were also detected but not reported, due to the inaccuracy in quantifying these elements via EDS.

The dimension of the as-printed samples was 4.44 cm (1.75″) long with a diameter of 0.635 cm (0.25″) and a final exposed surface area of 6.85 cm^2^. Before corrosion testing, the as-printed surface of the LPBF samples was removed through mechanical grinding, from 60 grit to 320 grit in successive steps. Immediately before immersion, air-formed films were removed by polishing the surface with 600-grit silicon carbide paper and then ultrasonically cleaning in ethanol for 5 min. After ultrasonically cleaning, the samples were dried with nitrogen and immediately immersed in the testing solution.

Localized corrosion resistance was measured through AC and DC electrochemical techniques. DC electrochemical techniques included open-circuit potential (OCP) and cyclic potentiodynamic polarization (CPP). AC electrochemical tests such as electrochemical impedance spectroscopy (EIS) and Mott–Schottky (MS) plots were used for characterization of the formed passive layer. All electrochemical tests were performed on a Gamry reference 600+ potentiostat™ (Gamry Instruments, Warminster, PA, USA). A conventional three-electrode setup was used with the LPBF 316L samples as the working electrode (WE), platinum mesh as the counter electrode, and tungsten oxide (W/Woxide) electrode as the reference electrode. The fabrication details of the W/Woxide and the conversion of the W/Woxide potentials to saturated calomel electrode (SCE) are detailed in the work by Cheng et al. [20]. A 3.5% NaCl solution with a buffered pH of 8 was used for corrosion testing. The pH of the solution was held constant through the use of borate buffer solution [8].

Electrochemical testing was performed in a glass cell held at a constant temperature using a recirculating heater. A condenser was used to keep evaporation to a minimum during testing. Once the solution was made and placed in testing cell, nitrogen was bubbled throughout the solution during testing to keep oxygen levels to a minimum. Before each electrochemical test, samples were immersed in the solution for up to 3 h to allow the surface film to stabilize before electrochemical testing. The CPP plots were produced by polarizing the WE from −800 mV vs. SCE to 1200 mV vs. SCE with the current limit set to be 1.20 mA/cm^2^. After the apex voltage/current was reached, the scan reversed until −100 mV vs. OCP. The scan rate for all tests was 0.1 mV/s. EIS measurements were performed at OCP with 10 mV_rms_ sinusoidal perturbation with frequencies from 100 kHz to 10 mHz. MS curves were produced at a frequency of 1 kHz with a potential range of −1000 mV to 1200 mV vs. SCE. The acceptor and donor densities were calculated from the following equations [7]:(1)$$\frac{1}{C^{2}}=\frac{2}{\epsilon \epsilon_{0}{eN}_{D}A^{2}}(V-V_{FB}-\frac{k_{B}T}{e})$$
(2)$$\frac{1}{C^{2}}=-\frac{2}{\epsilon \epsilon_{0}N_{A}A^{2}}(V-V_{FB}-\frac{k_{B}T}{e})$$
where ϵ is the relative dielectric constant, usually taken as 15.6 for stainless steels [21,22], $\epsilon_{0}$ is the permittivity of free space (F/m), *A* is the exposed surface area (m^2^), *V* is the applied potential (*V* vs. SCE), VFB is the flat band potential (*V* vs. SCE), $k_{B}$ is the Boltzmann constant (J/K), *T* is the temperature (K), and e is the charge of an electron (*C*). The acceptor and donor densities were calculated by finding the slope of the linear portions of the MS curves.

For the initial microstructural characterization, the samples were sectioned along the build direction and ground, from 240 grit silicon carbide paper to 1200 grit. Final polishing was performed using napped cloth with a 9 µm, 3 μm, and 1 μm diamond suspension solution and finished using a 0.04 μm SiO_2_ suspension solution. The samples were etched using aqua regia (3:1 HCl:HNO_3_) and the microstructure examined under a Nikon Eclipse MA 100 inverted microscope and a combination of TESCAN FERA3 and JEOL JCM-6000Plus SEM (JEOL USA, Inc., Peabody, MA, USA) for comparison of the as-printed microstructure versus its wrought counterpart. LPBF samples were also prepared for electron backscattered diffraction (EBSD) to obtain orientation information. EBSD was performed using the Oxford instruments detector (Oxford Instruments, Abingdon, UK) equipped in the TESCAN FERA3 SEM, and post-processing analysis was performed using the EBSD processing software from Oxford Instruments, AZtecCrystal software (Version 3.1). Additionally, focused ion beam (FIB) lift-out LPBF samples (~100 nm thickness) were prepared using an FEI Helios NanoLab 650 SEM (Thermo Fisher Scientific, Waltham, MA, USA) to expose the surfaces along the build direction to perform scanning transmission electron microscopy (STEM) analysis. STEM imaging and EDS were performed using JEOL 2800 STEM equipment, (JEOL USA, Inc., Peabody, MA, USA) at 200 keV accelerating voltage. X-ray photoelectron spectroscopy (XPS) was performed using an EnviroESCA system, which utilized a Mg X-ray source to characterize the passive films formed upon exposure to the test solution at various temperatures. The passive films were formed under open-circuit conditions by immersing the samples for 24 h; after immersion, samples were cleaned with DI water and dried with nitrogen. Immediately after cleaning and drying, the samples were placed in a desiccator and taken for XPS analysis. After a short sputtering time to remove any possible contaminations, a survey scan and individual scans for Cr, Fe, Ni, C, and O were performed. The XPS fitting was performed using the XPSPEAK41 software (https://public.wsu.edu/~pchemlab/index.html, accessed on 5 August 2023). The surfaces after CPP tests were examined under the SEM to study how the pit morphologies evolved with increasing test temperatures.

## 3. Results

### 3.1. Microstructure

Figure 1 shows the optical microscopy (OM) and scanning electron microscopy (SEM) images of the LPBF samples. Because the surface tested was along the build direction, the presence of semi-elliptical melt-pool boundaries (Figure 1a,b) can be observed in the direction opposite to the build direction. Figure 1c shows the heterogenous nature of the cell morphologies formed on the surface. A mixture of equiaxed, columnar, and mixed cells were observed at various regions, depending on the nature of cooling rates and degrees of undercooling. Figure 2a shows the microstructure of the wrought 316L sample, while Figure 2b,c show the magnified images of the LPBF in order to highlight the differences in the sizes and morphologies of cells/grains formed as well as the different features.

As can be observed, the wrought microstructure consisted of the typical grains and grain boundaries as well as some globular inclusions across the grains that were confirmed to be MnS inclusions from the SEM–energy dispersive spectroscopy (EDS) line scans. In comparison with this, the cellular structures formed in the LPBF (Figure 2b) sample were much finer and enclosed in cell boundaries. The cell widths ranged from about 0.2 to 2 μm, while the grain size of the wrought sample was about 10–35 μm. The high solidification rates occurring during the LPBF process were responsible for the formation of such fine structures, which were about two orders of magnitude lower than those of the wrought sample. Additionally, from the TEM image of the cell boundaries, it can be seen that there was a high density of dislocation networks at these regions. The possible reason for such dislocation networks has been attributed to the high localized strains occurring during the rapid cooling occurring during the LPBF process. The LPBF sample also showed the presence of some nano-inclusions, which were further analyzed using TEM–EDS maps (shown in Figure 3) and found to possibly be a mixture of Si and Cr oxides, which is similar to the results observed in a previous study on the same material [23]. Therefore, the anodic MnS inclusions present in the wrought SS316L sample were not present in the LPBF sample, which resolves the issues with pit formation occurring due to MnS inclusion dissolution. Scanning Kelvin probe force microscopy (SKPFM) maps of the LPBF samples reported in the literature indicate that the cell boundaries are at a higher Volta potential than the cell interiors, with the inclusions not indicating any contribution to the potential distribution [23]. Our previous characterization of this sample showed that Cr and Mo were enriched at the cell boundaries and depleted at the cell interiors and melt-pool boundaries, which can also be expected to influence the corrosion resistance of the LPBF samples in the environments tested in this study [23].

Electron backscattered diffraction (EBSD) maps obtained for the LPBF samples are shown in Figure 4a, with the presence of some melt tracks. Grain boundaries were detected during the post-processing of the EBSD data by setting the value higher than 5°, with about 68.3% of the boundaries being higher than 15°. Therefore, the majority of the boundaries were high-angle grain boundaries (HAGBs). The calculated misorientation angle distribution is presented in Figure 4b. It can be seen that majority of the misorientation angles ranged from about 20 to 50°. The scanning strategy used to manufacture the samples has a huge impact on the orientation. Changing the scanning strategy changes the temperature gradients and heat transfer characteristics, which in turn influences the final microstructure and crystallographic orientations [24]. The scanning strategy used in this study was a 90° rotation between two subsequently deposited layers, which can generally produce dense, relatively defect-free samples. For the parameters used to manufacture the samples used in this study, a mixture of various orientations was observed to be present.

### 3.2. DC Electrochemical Testing

Figure 5 compares the CPP curves of the wrought and LPBF samples at 25 °C, 50 °C, and 75 °C. The measured corrosion current densities (*i_corr_*) obtained from the polarization curves were similar in magnitude, around 10^−4^ mA/cm^2^ for all samples over all testing temperatures. This shows that initially, all samples displayed very low corrosion rates regardless of manufacturing method. Table 2 shows the OCP, breakdown potential (*E_B_*), repassivation potential (*E_RP_*), pitting susceptibility factor (*PSF*), and cumulative charge accrued after passive film breakdown (*C_B_*). The pitting susceptibility factor was calculated using Equation (3):(3)$$PSF=\frac{\left(E_{B}-E_{RP}\right)}{\left(E_{B}-E_{corr}\right)}$$

*PSF* values have been shown to help characterize both the pitting corrosion resistance and repassivation ability of stainless steels [25]. When *PSF* values are greater than 1 (*E_RP_* < OCP) this indicates that the material is shown to be highly likely to be susceptible to pitting corrosion at all potentials [26]. The *E_B_* for the LPBF samples were higher compared with those of the wrought samples at all testing temperatures. On average, the breakdown potential was around 120 mV more positive for the LPBF samples compared with the wrought, showing that the LPBF samples had a higher resistance to formation and propagation of stable pits at each tested temperature. *E_corr_* at each tested temperature for the wrought and LPBF samples were within 30 mV of each other. The measured *E_RP_* for wrought and LPBF decreased with increasing temperature.

The *PSF* for the wrought samples increased with increasing temperature, while that of the LPBF samples decreased. At 25 °C and 75 °C, the LPBF samples displayed higher *PSF* factors along with higher average *PSF* values over all temperatures compared with the wrought samples. Even with the higher pitting potentials of the LPBF samples, the PSF was closer to 1 due to the relatively lower *E_RP_*. After the passive film breakdown, the *C_B_* of the LPBF samples was higher than that of the wrought at all temperatures. The overall higher *PSF* values show that the LPBF samples are more susceptible to stable pitting over a larger range of potentials.

### 3.3. AC Electrochemical Testing

Figure 6 compares the Nyquist, Bode, and phase curves of wrought and LPBF samples at all three temperatures. At 25 °C and 50 °C, the phase angle for both wrought and LPBF samples is greater than −75° in the medium to low frequency range (10^−1^–10^0^ Hz), which is indicative of a compact and protective passive film. At 75 °C, the phase angle was shifted to more positive values between −60° and −70°, with this possibly due to the formation of a more defective passive film. It can be seen from the Bode and phase-angle plot that at temperatures above 25 °C, the LPBF samples had a more negative phase angle and a higher impedance magnitude at the lower end of the frequency spectrum (<10^−1^ Hz). At higher temperatures, the passive film formed on the LPBF samples is providing more protection compared with its wrought counterpart. The used equivalent electrical circuit consists of two parallel resistance–constant phase element (R-CPE) circuits connected in series with each other and with the solution resistance. This circuit is commonly used for fitting passive films formed on stainless steel [27,28]. Table 3 shows the fitting values obtained from fitting the equivalent circuit to the EIS data. CPE circuits were chosen due to the non-ideal capacitive behavior of the passive film and double layer. CPE impedance is calculated using Equation (4):
(4)$$Z_{CPE}=\frac{1}{Q{\left(\omega i\right)}^{n}}$$
where *Q* is a constant with units of $\frac{s^{n}}{\Omega }$, ω is the angular frequency with units of Hz, and *n* is a dimensionless value representing the behavior of the *CPE*. The value of *n* varies between 0 and 1, with 0 meaning the *CPE* is a perfect resistor and 1 a perfect capacitor. The effective capacitance was calculated from the *CPE* terms using the model proposed by Hsu and Mansfeld [29]. $C_{eff}$ for the film was calculated using Equation (5):(5)$$C_{eff,f}=Q_{\;f}^{1/n_{f}}R_{f}^{(1-n_{f})/n_{f}}$$
where the effective capacitance for each *R*-*CPE* circuit is calculated with the associated *R* and *CPE* parameters associated with the film.

At 25 °C, the passive film formed on both wrought and LPBF samples showed similar levels of protection from the fitted values, but with increasing temperature, the film formed on the LPBF samples was more protective compared with the wrought samples. For both wrought and LPBF samples, with increasing temperature the *R_f_* and *R_ct_* values decreased and *Q_f_*, *Q_ct_*, and *C_f_* increased. The increase in these values along with the decrease in *n_f_* and *n_dl_* shows that the film formed at higher temperatures has increased surface heterogeneity and defect density.

The Mott–Schottky plots of the formed passive films at various temperatures for the samples are shown in Figure 7.

Portions of the graph with a positive slope indicate n-type semiconductor properties, while negative slopes are indicative of p-type semiconductor properties [30]. At each temperature, both 170 W and wrought samples displayed a turning point where the passive film switched from n-type semiconductor to p-type semiconductor. The calculated acceptor and donor defect densities are shown in Table 4. The turning point of the MS curves showed that the passive film formed at more negative potentials was mainly composed of n-type components: Fe_2_O_3_, MoO_3_, and FeOOH. Potentials above the turning point of the passive film consisted mainly of p-type components: Cr_2_O_3_, MoO_2_, FeCr_2_O_4_, and NiO [7]. At each temperature, the number of defects calculated in the wrought samples was higher than that of the 170 W samples.

The lower number of defects in the LPBF film could possibly explain why it displayed a higher initial protection to the starting of the corrosion process compared with the wrought samples. With increasing temperature, both donor and acceptor defect densities for both wrought and LPBF samples increased.

### 3.4. XPS Measurements

After one day of immersion in the buffered 3.5 wt% NaCl solution, XPS was performed to understand the composition of the passive film formed at each test temperature. A survey scan was first performed to identify the elements present, after which individual scans were performed for the elements. Figure 8 and Figure 9 show the detailed spectra of Cr 2p_3/2_, Fe 2p_3/2_, and Ni 2p_3/2_ in the passive film for the LPBF and wrought samples at 50 and 75 °C. XPS was also performed for the 25 °C sample as a control, and the spectra are presented in the Appendix A.

For the Fe 2p_3/2_ spectrum, XPS fitting revealed the presence of Fe metal (706.5 eV) [31], Fe_3_O_4_ (709.2 eV) [32], and FeOOH (711 eV) [33]. From the Cr 2p_3/2_ spectrum, the presence of Cr metal (573.9 eV) [34], Cr_2_O_3_ (576.1 eV) [35], and Cr(OH)_3_ (577 eV) [36] was revealed. The Mo 3d spectrum was fit with two peaks each of Mo metal (227.4 eV, 231 eV) [37,38], MoO_2_ (230 eV, 234.2 eV) [37,39], and MoO_3_ (232.3 eV, 236 eV) [40,41] for the 3d_5/2_ and 3d_3/2_ peaks. The Ni 2p_3/2_ peaks were fit with Ni metal [42] and NiO/Ni(OH)_2_ [43]. Therefore, the primary components of the passive film for both the samples at all the tested temperatures were a mixture of Fe, Cr, Mo, and Ni oxides, which are typical of stainless steel 316L. The binding energies and amounts of the components from the individual spectra calculated using the CasaXPS software (Version 2.3.17) for the LPBF and wrought samples are presented in Table 5. Based on the individual amounts of the elements calculated from the survey scan, the overall amounts of oxides and hydroxides in the passive film were calculated and are presented as bar graphs in Figure 10.

The overall amounts of Fe and Cr were higher on the outer layer of the passive film for the LPBF samples than the wrought samples at all temperatures, possibly due to the dense dislocation networks that are typically present at the cell boundaries in the LPBF samples, which serve as diffusion pathways [23,44]. In general, the amounts of hydroxide components were found to increase in comparison with the oxide components with increasing temperatures. The metallic components of Fe and Cr generally decreased with increasing temperatures due to the formation of a thicker passive film, as supported by the capacitance values from the EIS results. Looking at the amounts of Cr_2_O_3_ and Cr(OH)_3_ present on the films formed on the LPBF samples, the amounts were generally higher than those formed on the wrought samples at all tested temperatures, which is possibly the reason for improved corrosion resistance and passive film stability. The amounts of Fe_3_O_4_ and FeOOH were higher in the LPBF samples as well. The amounts of Mo oxides were similar in both the wrought and LPBF samples. With increasing temperatures, the amount of Cr and Fe content in the outer layer of the passive film decreased, which was possibly the reason for the lowering corrosion resistance.

## 4. Discussion

The LPBF samples definitively showed a higher resistance to the initiation of pitting compared with conventionally manufactured components over all temperatures. The increased passivation capability of the LPBF samples was possibly due to the increased dislocation density at the cell boundaries and smaller cell sizes, which can provide pathways for the diffusion of Cr and Mo to the surface to form a stable, compact, and dense passive film. However, the overall corrosion resistance of the LPBF samples is not as definitive due to a larger range of metastable pitting and much larger hysteresis loops. The larger hysteresis loops could be attributed to the formation of micro-galvanic cells due to the chemical segregation in the cell boundaries and melt-pool boundaries [23,45]. Once the passive layer on the LPBF-produced samples breaks down, allowing the corrosion process to occur freely, it is most likely to be concentrated at the galvanic couple boundary. Due to the very small cell size relative to the grain size of commercially produced components, when a high density of these micro-galvanic couples exists in a local area, it is possible that a larger accumulation of pitting damage could occur on the 170 W samples compared with what would occur on conventionally produced samples over the same time frame.

After the CPP tests at all the tested temperatures, the surfaces of the samples were cleaned thoroughly before performing scanning electron microscopy (SEM) to reveal the morphology of pits and possible microstructural features. The morphology of pits formed on LPBF and wrought samples have been documented well in many works [46,47,48], and hence, the results obtained in this study are presented in the Appendix A to serve as a control to compare the higher-temperature results against. Figure 11 and Figure 12 show the pits formed on the wrought 316L sample after CPP in 3.5 wt% NaCl at 50 and 75 °C, respectively.

The pit density was higher at 75 °C, while the pit morphologies looked similar at both the tested temperatures, with the widths ranging from 2 to 200 μm. The pit interiors showed the etched microstructure of the wrought sample (Figure 11a and Figure 12a) with evidence of attack at the triple junctions and grain boundaries (Figure 11b,c and Figure 12b,d). Additionally, there were some smaller pits and regions within the pit interiors that also showed evidence of dissolution of the MnS inclusions that were distributed across the grains of the wrought sample. Because these inclusions are anodic, they have generally been found to be preferred regions of attack in commercial stainless steels [49].

Figure 13 and Figure 14 show the pit morphologies on the surfaces of the LPBF samples after CPP at 50 and 75 °C, respectively. The pit morphologies were similar to those on the wrought samples and ranged from 5 to 300 μm. The pit interiors showed some partially dissolved underlying layers with an etched appearance. Magnified images of the pit interiors (Figure 13c,d and Figure 14c,d) showed the attack at the melt-pool boundaries as well as an etched appearance of the cells and cell boundaries due to the selective dissolution of cell interiors. This was in contrast to that observed in the wrought 316L sample, which mostly showed the attack originating from the MnS inclusions and the grain boundaries.

The occurrence of preferential attack at the melt-pool boundaries as well as the cell interiors has been well documented in other works in similar test environments [45,47,50,51,52]. This has usually been reported to be due to the micro-segregation of Cr/Mo to the cell boundaries, which caused them to be more cathodic or less prone to attack, while the interiors and melt-pool boundaries showed a depletion of these elements, which made them more anodic. Therefore, even though there was an absence of anodic MnS inclusions in the 170 W sample, the heterogenous distribution of key passivating elements such as Cr/Mo can lead to local heterogeneities in the passive film formed as well as contribute to the preferential attack of specific features. The high degree of metastable pitting observed in the CPP results could also be attributed to these local differences in the passive film formed. Therefore, even at higher temperatures, the preferred sites of localized attack remained the melt-pool boundaries and cell interiors.

From AC electrochemical testing and XPS results, it can be seen that the protectiveness of the formed passive film for both wrought and LPBF 316L samples decreases with increasing formation temperature. At each formation temperature, the formed passive film on the LPBF 316L was more protective compared with its wrought counterpart. With increasing temperatures, the effectiveness of the formed passive film decreased due to the formation of thicker, defective passive films. This increase in defect density was confirmed through MS testing, with the defect density increasing by an order of magnitude (10^19^ → 10^20^) for films formed at 25 °C compared with 75 °C. The decreasing effectiveness of the passive film formed on the LPBF samples at higher temperatures still provided more protection to the initiation of localized corrosion compared with the wrought samples. From EIS testing, the impedance of the samples showed a decrease in the *R_f_* and *n_f_,* with an increase in *Q_f_* and *C_f_* values, with increasing temperatures. The shifts in these values with increasing temperature is indicative of the increasing heterogeneity and defectiveness of the formed passive film.

The overall protectiveness of the formed passive film on wrought and LPBF 316L was studied through the combination of electrochemical tests and surface characterization. Figure 15 shows the effect of relative composition and defect density on the effectiveness of the passive film formed on LPBF 316L.

With increasing temperatures, the resistance to localized corrosion decreases. This decrease is most likely due to the changing passive film composition and increased heterogeneity and defect density of the formed passive film.

## 5. Conclusions

The passive film and pitting characteristics of LPBF SS316L was compared with those of a commercially obtained wrought 316L at 25, 50, and 75 °C. Passive film performance and localized corrosion were measured with AC and DC electrochemical techniques and high-resolution analytical surface techniques. Based on the results obtained in this study, the following conclusions can be drawn:(i)Comparing the microstructure of LPBF 316L with wrought showed that the size of the features formed in the LPBF samples were about two orders of magnitude lower than those in the wrought samples, and they showed various morphologies, such as equiaxed and columnar. There was an absence of anodic MnS inclusions in the LPBF samples, with the presence of silicon/Cr oxide nano-inclusions distributed along the cell boundaries instead. Additionally, dislocation networks were found to be present at the cell boundaries.(ii)Cyclic potentiodynamic polarization showed that the 170 W samples showed a higher initial resistance to the onset of localized corrosion compared with wrought 316L. The LPBF samples were more susceptible to metastable pitting over a larger range of potentials compared with the wrought samples. But for the LPBF samples, once pitting has begun, the amount of accumulated damage was higher compared with the wrought 316L.(iii)EIS testing showed that the film formed at OCP on 170 W samples provided more protection compared with film formed on the wrought samples. Mott–Schottky analysis showed that both wrought and 170 W showed a transition of n-type to p-type around OCP. The calculated donor densities were similar for both wrought and LPBF 316L and increased with increasing formation temperature.(iv)XPS results revealed that the amounts of Cr and Fe oxides and hydroxides were higher on the passive films formed on the LPBF samples at all tested temperatures when compared with the wrought samples, which made the film more corrosion resistant and was the reason for its improved stability.(v)Pits formed on the LPBF showed attack on deposited layers underneath the tested surface once the top surface dissolved, with melt-pool boundaries and cell interiors being the preferential regions of attack due to being depleted in Cr and Mo.

Overall, the LPBF 316L samples were more resistant to the initiation of localized corrosion at all testing temperatures compared with the wrought samples. This increased resistance persisted through the detrimental effect of temperature on the protectiveness of the formed passive film. The increased corrosion resistance of the LPBF samples did not persist once the localized corrosion process occurred. After passive film breakdown, the formation of micro-galvanic cells between the Cr-rich cell boundaries and cell interiors was possibly responsible for the increased damage that occurred in LPBF 316L samples.

## Figures and Tables

**Figure 1 materials-17-03420-f001:**
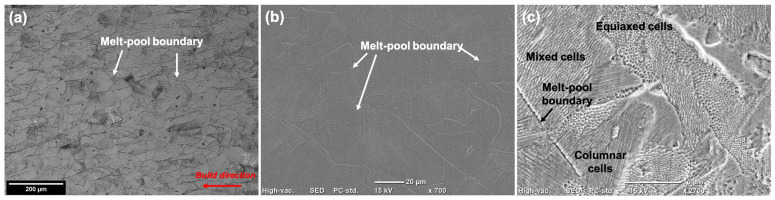
(**a**) Optical microscopy (OM) image; (**b**,**c**) scanning electron microscopy (SEM) images showing the semi-elliptical melt-pool boundaries and the heterogenous cell morphologies present in the LPBF samples.

**Figure 2 materials-17-03420-f002:**
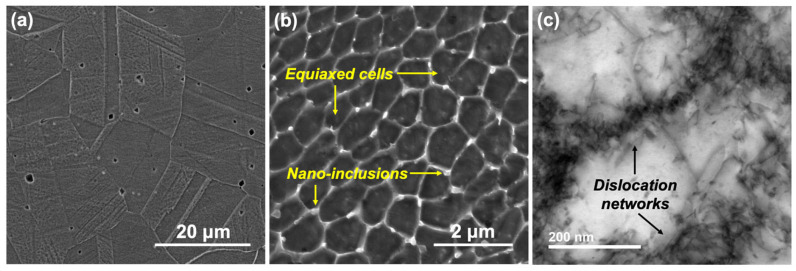
SEM images showing the microstructure of the (**a**) wrought 316L; (**b**) magnified images showing the equiaxed cells as wells as some nano-inclusions present in the LPBF 316L sample; (**c**) transmission electron microscopy (TEM) images showing the dislocation networks formed at the cell boundaries.

**Figure 3 materials-17-03420-f003:**
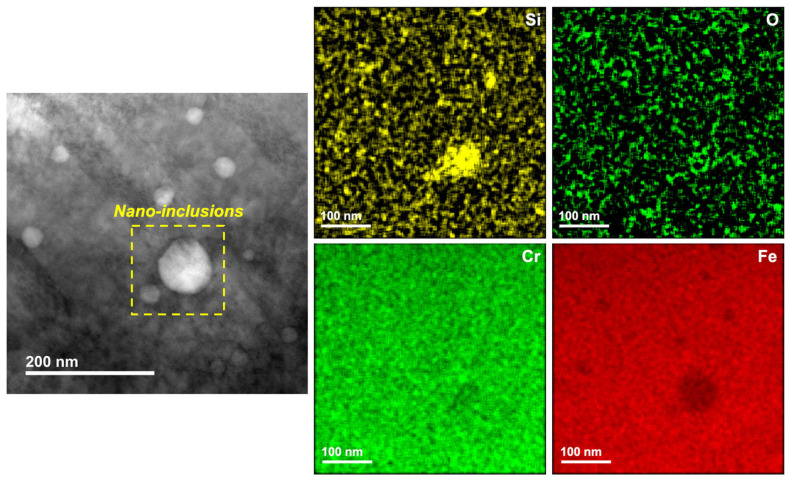
STEM–EDS maps showing the nature of nano-inclusions formed in the LPBF SS316L sample.

**Figure 4 materials-17-03420-f004:**
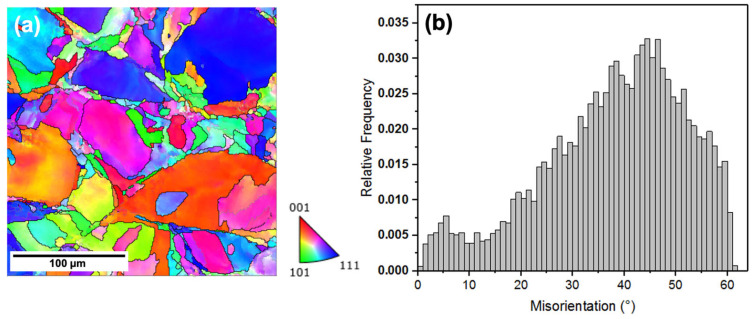
Initial characterization of the surface along the build direction of the LPBF sample using EBSD showing the (**a**) inverse pole figure (IPF) map and (**b**) misorientation angle distribution obtained after analysis.

**Figure 5 materials-17-03420-f005:**
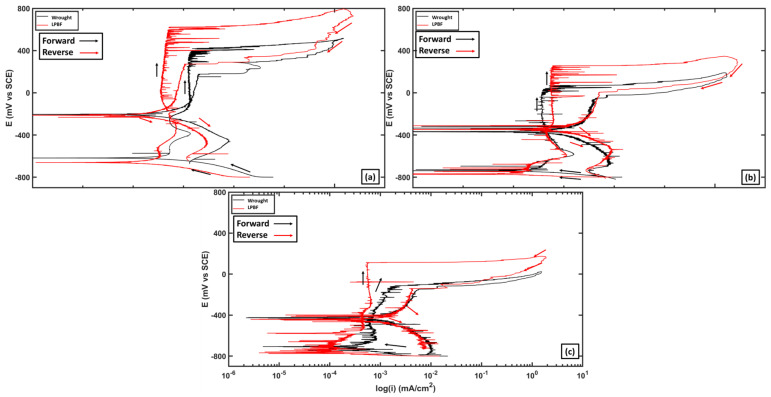
Comparison of the CPP curves of wrought and LPBF samples at (**a**) 25 °C, (**b**) 50 °C, and (**c**) 75 °C.

**Figure 6 materials-17-03420-f006:**
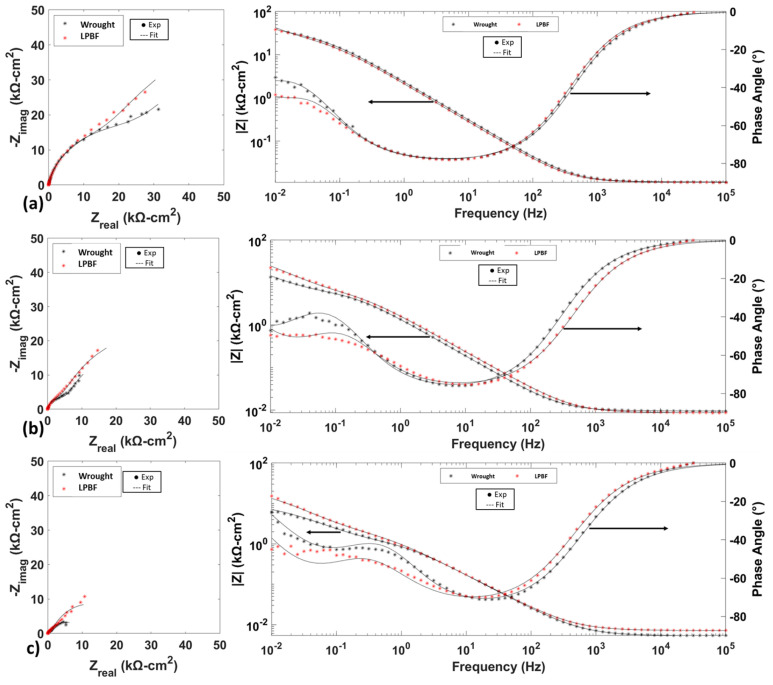
Nyquist, Bode, and phase-angle plots of wrought and LPBF 316L at (**a**) 25 °C, (**b**) 50 °C, and (**c**) 75 °C.

**Figure 7 materials-17-03420-f007:**
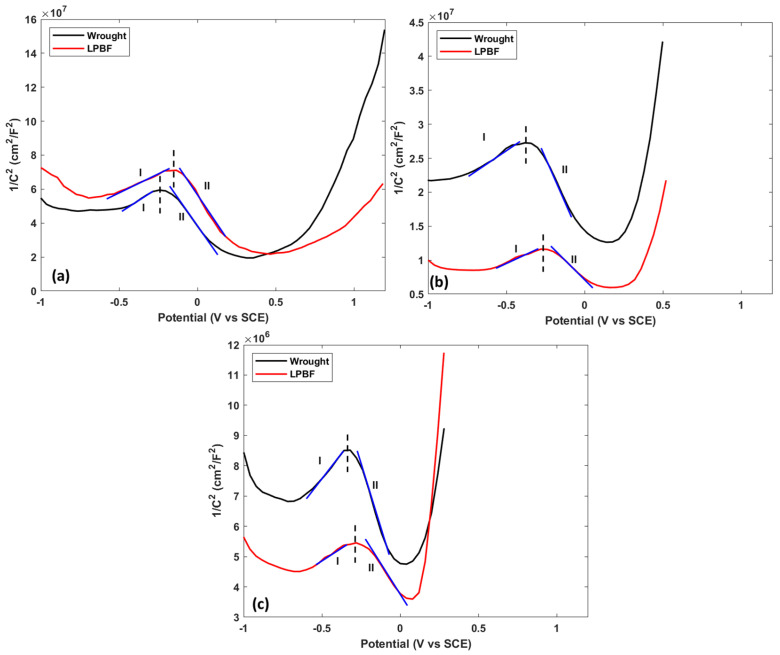
Mott–Schottky curves of wrought and LPBF samples at (**a**) 25 °C, (**b**) 50 °C, and (**c**) 75 °C. Blue lines indicate the lines used for the calculation of the slope, and I and II signify the regions where the material displays n-type and p-type semiconductor properties respectively.

**Figure 8 materials-17-03420-f008:**
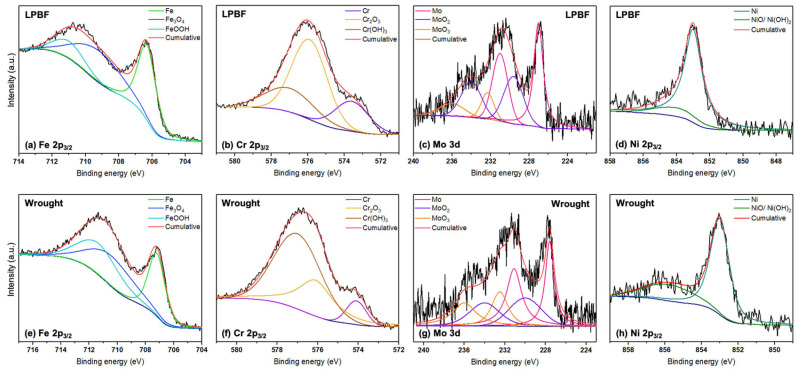
Individual high-resolution XPS spectra of the Fe 2p_3/2_, Cr 2p_3/2_, Mo 3d, and Ni 2p_3/2_ peaks obtained for the (**a**–**d**) LPBF and (**e**–**h**) wrought SS316L samples after 1 day of exposure in the buffered 3.5 wt% NaCl solution at 50 °C (The black lines represent the raw spectra).

**Figure 9 materials-17-03420-f009:**
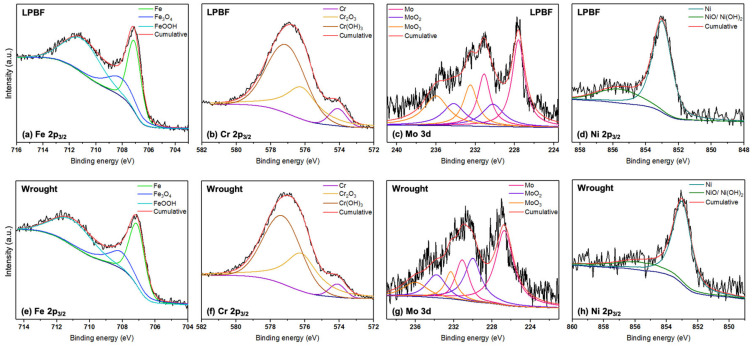
Individual high-resolution XPS spectra of the Fe 2p_3/2_, Cr 2p_3/2_, Mo 3d, and Ni 2p_3/2_ peaks obtained for the (**a**–**d**) LPBF and (**e**–**h**) wrought SS316L samples after 1 day of exposure in the buffered 3.5 wt% NaCl solution at 75 °C (The black lines represent the raw spectra).

**Figure 10 materials-17-03420-f010:**
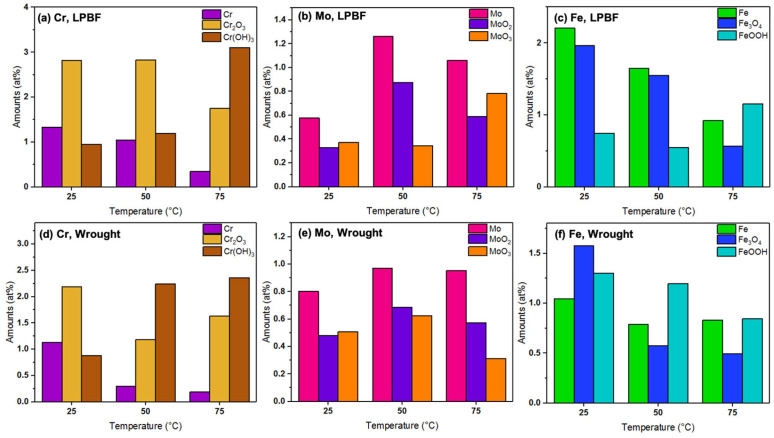
Amounts of Cr, Mo, and Fe components in the passive film formed on the (**a**–**c**) LPBF and (**d**–**f**) wrought SS316L samples.

**Figure 11 materials-17-03420-f011:**
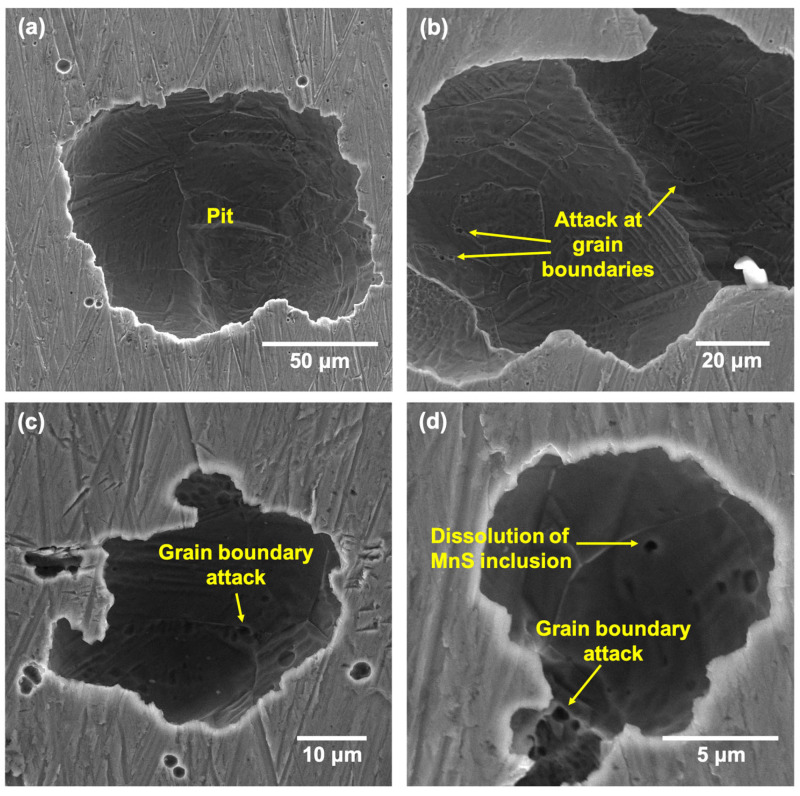
SEM images showing (**a**) the typical pit morphology formed on the wrought SS316L sample after CPP tests at 50 °C and (**b**–**d**) showing instances of MnS inclusion dissolution and grain boundary attack.

**Figure 12 materials-17-03420-f012:**
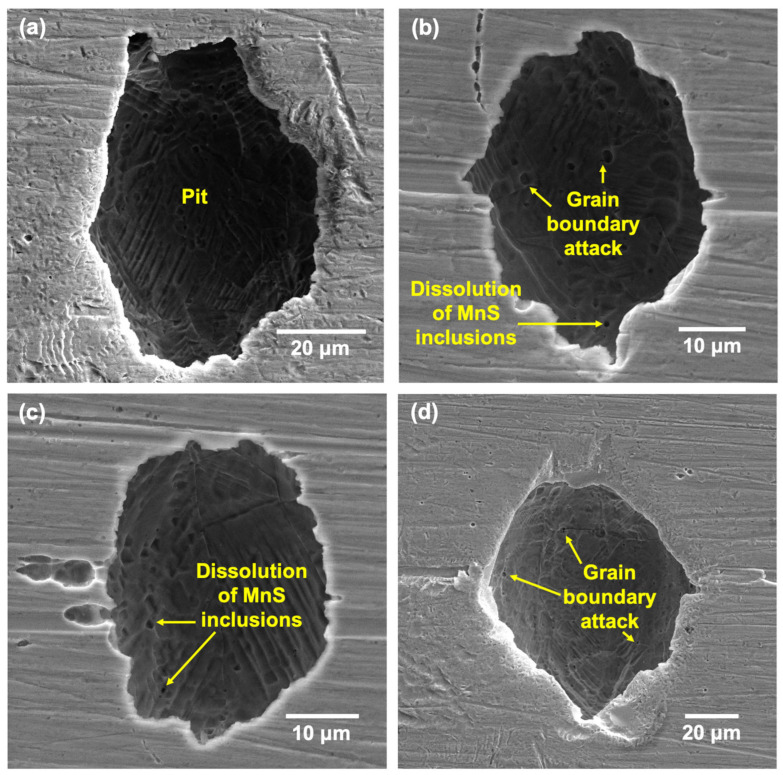
SEM images showing (**a**) the typical pit morphology formed on the wrought SS316L sample after CPP tests at 75 °C and (**b**–**d**) pits showing instances of MnS inclusion dissolution and grain boundary attack.

**Figure 13 materials-17-03420-f013:**
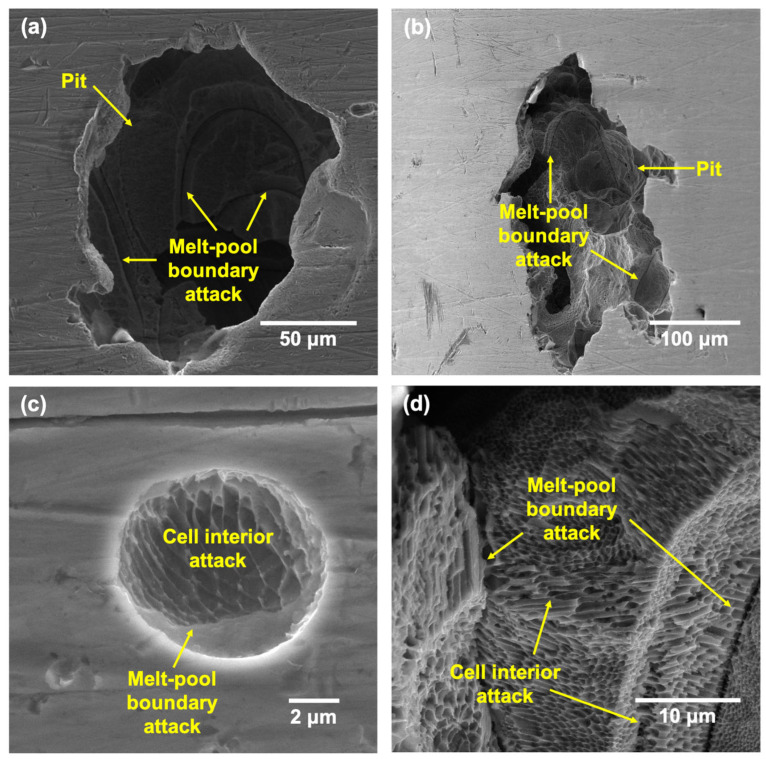
SEM images showing the (**a**,**b**) pits formed on the LPBF SS316L sample after CPP tests at 50 °C and (**c**,**d**) instances of cell interior dissolution and melt-pool boundary attack.

**Figure 14 materials-17-03420-f014:**
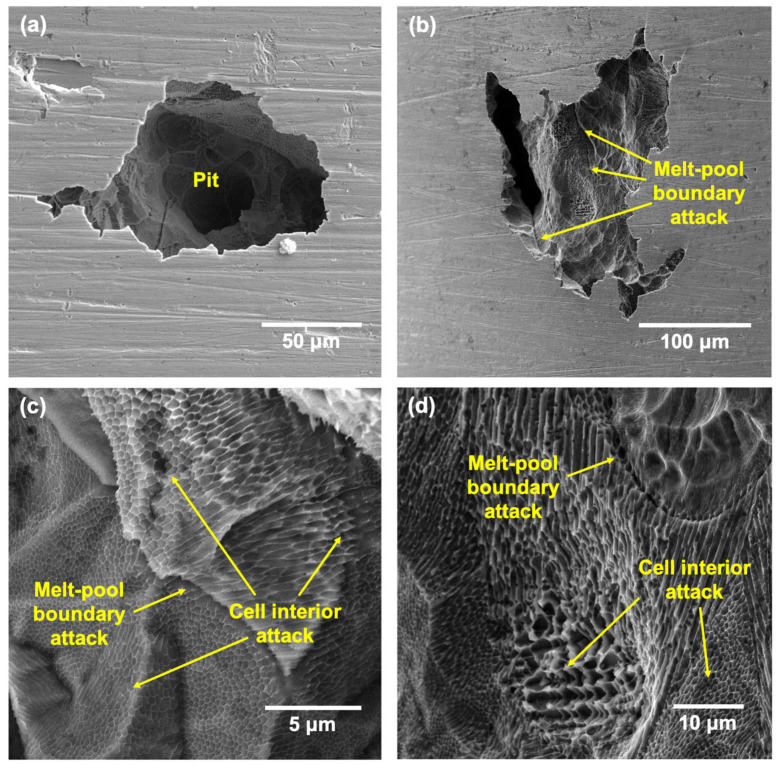
SEM images showing the (**a**,**b**) pits formed on the LPBF SS316L sample after CPP tests at 75 °C and (**c**,**d**) magnified images showing instances of cell interior dissolution and melt-pool boundary attack.

**Figure 15 materials-17-03420-f015:**
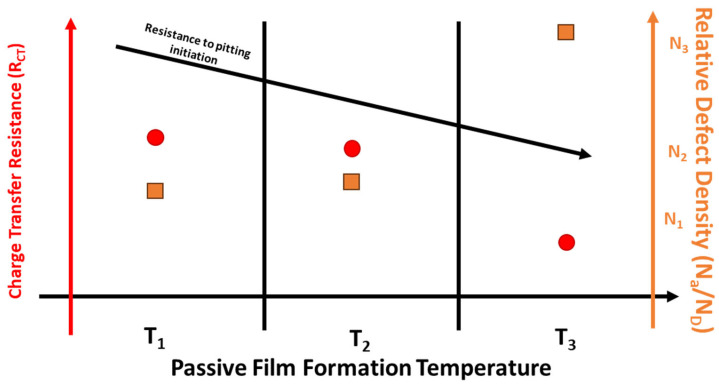
Relative passive film chromium amounts and defect density for LPBF 316L formed at various temperatures.

**Table 1 materials-17-03420-t001:** Average relative composition of LPBF and wrought 316L measured by EDS.

Sample	Fe (%)		Cr (%)		Ni (%)		Mo (%)		Mn (%)	
	Avg.	(SD)	Avg.	(SD)	Avg.	(SD)	Avg.	(SD)	Avg.	(SD)
Wrought	58.8	1.15	17.5	0.17	11.9	0.10	2.4	0.32	1.1	0.10
LPBF	58.7	0.35	17.3	0.15	12.8	0.30	2.4	0.22	1.5	0.21

**Table 2 materials-17-03420-t002:** Comparison of OCP, *E_B_*, *E_RP_*, *PSF*, and *C_B_* of wrought and LPBF samples.

Sample	Temperature (°C)	*E_corr_* (mV vs. SCE)	*E_B_* (mV vs. SCE)	*E_RP_* (mV vs. SCE)	*PSF*	*C_B_* (C/cm^2^)
Wrought	25	−610.08	425.69	−30.72	0.44	1.81
	50	−683.07	138.44	−320.65	0.56	2.44
	75	−708.55	−114.00	−402.87	0.49	1.69
LPBF	25	−609.31	585.07	−130.72	0.60	5.87
	50	−709.66	223.60	−263.63	0.52	5.66
	75	−737.89	42.53	−375.95	0.54	2.87

**Table 3 materials-17-03420-t003:** Equivalent circuit fitting values.

Sample	Temperature (°C)	*R_s_* (Ω-cm^2^)	*R_f_* (kΩ-cm^2^)	*R_ct_* (kΩ-cm^2^)	*Q_f_* (F-cm^−2^-s^−n^)	*n_f_*	*C_f_* (F-cm^−2^-s^−n^)	*Q_dl_* (F-cm^−2^-s^−n^)	*n_dl_*
Wrought	25	12.49	24.41	62.01	1.11 × 10^−4^	0.88	1.28 × 10^−4^	5.47 × 10^−4^	0.88
	50	9.37	6.36	80.25	1.55 × 10^−4^	0.84	1.54 × 10^−4^	5.75 × 10^−4^	0.88
	75	5.31	0.77	8.56	2.18 × 10^−4^	0.77	1.76 × 10^−4^	7.12 × 10^−4^	0.89
LPBF	25	14.32	25.59	74.03	1.28 × 10^−4^	0.87	1.53 × 10^−4^	4.52 × 10^−4^	0.95
	50	10.59	7.30	71.40	1.57 × 10^−4^	0.85	1.61 × 10^−4^	4.67 × 10^−4^	0.88
	75	7.84	1.17	20.85	2.57 × 10^−4^	0.81	2.12 × 10^−4^	4.61 × 10^−4^	0.86

**Table 4 materials-17-03420-t004:** Calculated N_A_ and N_D_ from MS plots.

Sample	Temperature (°C)	N_A_ (#/cm^3^) ^a^	N_D_ (#/cm^3^) ^a^
Wrought	25	5.50 × 10^19^	2.67 × 10^19^
	50	4.33 × 10^20^	2.02 × 10^20^
	75	1.12 × 10^21^	5.23 × 10^20^
LPBF	25	3.76 × 10^19^	1.80 × 10^19^
	50	1.18 × 10^20^	6.27 × 10^19^
	75	7.10 × 10^20^	2.47 × 10^20^

^a^ the number of defects calculated per volume.

**Table 5 materials-17-03420-t005:** XPS peak fitting parameters for the elements and amount calculated from each spectrum for the LPBF and wrought samples at the various test temperatures.

Samples:			LPBF			Wrought		
Element	Peak	Binding Energy (eV)	25 °C (at%)	50 °C (at%)	75 °C (at%)	25 °C (at%)	50 °C (at%)	75 °C (at%)
Fe 2p_3/2_	Fe metal	706.5	44.9	44.0	35	26.6	30.8	38.3
	Fe_3_O_4_	709.2	39.9	41.4	21.4	40.2	22.5	22.8
	FeOOH	711	15.2	14.6	43.6	33.2	46.7	38.9
Cr 2p_3/2_	Cr metal	573.9	26.1	20.6	6.6	26.9	8.1	4.6
	Cr_2_O_3_	576.1	55.3	55.9	33.7	52.2	31.2	39.0
	Cr(OH)_3_	577	18.6	23.5	59.7	20.9	60.7	56.4
Mo 3d	Mo 3d_5/2_ Mo 3d_3/2_ MoO_2 _3d_5/2_ MoO_2_ 3d_3/2_ MoO_3_ 3d_5/2_ MoO_3_ 3d_3/2_	227.4 231.0 230 234.2 232.3 236	45.2 25.6 29.2	50.8 35.3 13.9	43.6 24.2 32.2	44.8 26.9 28.3	42.6 30.1 27.3	51.7 31.2 17.1
Ni 2p_3/2_	Ni metal	853	85.1	86.9	76.7	71.2	73.2	75.3
	NiO/Ni(OH)_2_	855.6	14.9	13.1	23.3	28.8	26.8	24.7

## Data Availability

The data presented in this study are available on request from the corresponding author due to the results being a part of an ongoing study.

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
