# Peer review of "Effect of Temperature on Passive Film Characteristics of LPBF (Laser Powder-Bed Fusion) Processing on UNS-S31603"

_materials, 2024, doi:10.3390/ma17143420_

Round 1

Reviewer 1 Report

Comments and Suggestions for Authors

The manuscript titled: "Effect of temperature on passive film characteristics of LPBF (Laser Powder Bed Fusion) processing on UNS-S31603'' describes the influence of temperature on the initial corrosion of 316L steel produced by LPBF. The authors describe the corrosion studies in detail and accurately, but the correlation with the initial microstructure raises some concerns. After revisions and supplementation of the microscopy section, I recommend accepting the article for publication. Detailed comments:"

1.      "44-47: 'The microstructure of LPBF components is characterized by a cellular-like sub-grain structure perpendicular to the build direction and a columnar sub-grain structure parallel to the build direction [13-15].'

I disagree with this – the orientation of the grain and cellular substructure depends on many parameters. It is true that there is a preferred growth direction in the material, but it can be disrupted by many factors."

2.      "Are there any studies in the literature where LPBF 316L steel exhibits worse corrosion resistance compared to conventional alloys? If so, what are the reasons for this?"

3.      The description of the microstructure lacks essential information. There are no SEM images at lower magnifications that would show the morphology and grain size. The microstructure after LPBF is multi-scale, and focusing only on the smallest features can lead to errors.

a.      "Which plane of the sample was examined?

b.      What was the grain size?;

c.      Was there a dominant orientation in the material?;

d.      Are the 'inclusions' marked in Figure 1 not primary precipitates? If so, what type are they?

e.      Were there any oxide inclusions in the material?

f.        What was the shape and character of the meltpool?

g.       Was the porosity measured?; What type was it?"

4.      A note for the future: Conducting studies on polished samples would reveal a much larger area for further SEM investigation."

5.      Are the authors sure that corrosion initiation did not occur in the pore areas?

6.      Is Cr segregation to cell walls really observed?

Reviewer 2 Report

Comments and Suggestions for Authors

The paper presents a corrosion study comparing AM and wrought 316L. The paper is well-written and clear. I only have a few comments.

Table 1 shows the nominal composition. I believe some of the values are max values (C, Mn, Si etc.). It would be good to state if this is the case. It would also be valuable to know the actual compositions of the two materials, as differences in the composition could be partly responsible for the differences in corrosion properties.

From the XPS results it is inferred that the oxides consist of Cr2O3, Cr(OH)3, etc. Do you mean that the oxide consists of many phases (crystalline or amorphous?), or is it one phase consisting of several elements? In my opinion TEM/EDS often show that the oxides are mixed (say M2O3). Of course, also a mixed oxide has a lot of bonds like Cr-O, Fe-O, etc. It would have been interesting to also characterize the oxide by TEM (or APT), but that is of course a lot of work, and maybe the oxides are so thin that even such techniques have problems to resolve the true nature of the oxide. Maybe grazing incidence XRD could reveal the crystal structure? I do not request more experiments to be done, but the nature of the oxide could be discussed.

On line 240, the reference to Table 4 is missing.

The text on line 251-255 is the same as on line 240-245.

In table 5, four significant digits are used for the various components. Does the method used motivate this high precision? I think two (or maybe three) significant digits would more reflect the actual precision.

Round 2

Reviewer 1 Report

Comments and Suggestions for Authors

Thank you very much for adding the part concerning microstructure analysis. In the corrected part of the manuscript you sent, there are still some errors. Please address the following comments:

  1. In the research description, there is a lack of specification of individual methods. If the chemical composition was determined using EDS, the microscope parameters and the area of the studied region are crucial. What camera and software were used for EBSD studies? At what parameters were STEM studies conducted?, etc.
  2. In quantitative EDS measurements, usually the content of light elements is not provided as they are associated with a very high error. In Table 2, for the nominal composition, the value of C at 7% is several times higher than it should be. I suggest not providing the content of these elements or determining them using other spectroscopic methods.
  3. If average values are given in the table, what was the standard deviation of the results? In Figure 4, IPF maps for three different vectors are shown. Repetition of this does not bring any new information. In the case of LPBF samples, they should be oriented towards the building direction and/or scanning direction. If the authors refer to grain boundary misorientation, it is good to present a histogram of their distribution.
  4. In Line 201-202: "A general mixture of various orientations was observed which is characteristic of the scanning strategy used which in this study is ± 45° which is a 90° rotation between two subsequently deposited layers." What was the rotation strategy? And what effect does it have on corrosion?
  5. In my opinion, the EDS maps shown in Figure 3 were conducted in scanning-transmission , not transmission.
  6. In the description of figure 4, the reference to the surface may be confusing. Is it referring to the sample surface? If so, in which orientation? Or is it about the surface of corrosion tests?"
